# Sarcopenia with Depression Presents a More Severe Disability Than Only Sarcopenia among Japanese Older Adults in Need of Long-Term Care

**DOI:** 10.3390/medicina59061095

**Published:** 2023-06-06

**Authors:** Takahiro Shiba, Ryo Sato, Yohei Sawaya, Tamaki Hirose, Masahiro Ishizaka, Akira Kubo, Tomohiko Urano

**Affiliations:** 1Nishinasuno General Home Care Center, Department of Day Rehabilitation, Care Facility for the Elderly “Maronie-en,” 533-11 Iguchi, Nasushiobara 329-2763, Tochigi, Japan; 2Department of Physical Therapy, School of Health Sciences, International University of Health and Welfare, 2600-1 Kitakanemaru, Otawara 324-8501, Tochigi, Japan; 3Department of Geriatric Medicine, School of Medicine, International University of Health and Welfare, 4-3 Kozunomori, Narita 286-8686, Chiba, Japan

**Keywords:** sarcopenia with depression, long-term care certification level, physical function, nutrition, older adults

## Abstract

*Background and objectives*: The combination of depression and sarcopenia, a condition prevalent worldwide, may cause dis-tinct problems that should not be underestimated. However, to the best of our knowledge, no reports have investigated the combined effects of depression and sarcopenia. In this study, we compared physical function, nutritional status, and daily functioning among older adults with only depression (OD), those with only sarcopenia (OS), and those with sarcopenia with depression (SD) to examine the effects of the combination of depression and sarcopenia. *Materials and Methods*: The participants were 186 communi-ty-dwelling older individuals who required support or care. The participants were classified into four groups based on the presence or absence of sarcopenia and depression: Control, OD, OS, and SD. The following parameters were evaluated in the four groups: grip strength, walking speed, skeletal muscle mass index (SMI), Mini Nutritional Assessment Short-Form (MNA-sf), and long-term care certification level. In addition, univariate and multivariate analyses of the survey results were performed to identify risk factors leading from OS to SD. *Results*: We found that 31.2% of older participants who needed support or nursing care had SD, which had more pronounced adverse effects on grip strength, walking speed, SMI, MNA-sf, and level of nursing care than OD or OS. Furthermore, multivariate analysis of SD vs. OS showed that decreased grip strength and worsening MNA-sf were independent risk factors. *Conclusions*: SD is common among older individuals living in the community. Patients with SD require support and care, and the condition has a greater impact on physical function, nutritional status, and decline in life function compared to OD and OS. Therefore, it is desirable to elucidate the process leading to SD and investigate the risk factors and prognosis. It is expected that sarcopenia with depression will be investigated worldwide in the future.

## 1. Introduction

Sarcopenia is characterized by the progressive and generalized loss of skeletal muscle mass and strength with age and disease [1]. Sarcopenia contributes not only to the decline in physical function but also to a higher probability of falls and fractures [2], worsening cardiopulmonary function [3,4], diminished activities of daily living (ADL) and quality of life [5,6,7], and increased mortality [8,9]. Epidemiological studies reported that more than 50 million people worldwide were affected by sarcopenia in 2016, and this number may increase to 200 million in the next 40 years [10]. Research and studies on sarcopenia are being conducted around the world to extend the healthy life expectancy of older adults.

In recent years, new concepts of sarcopenia have been reported, such as osteosarcopenia, associated with osteoporosis [11,12]; respiratory sarcopenia, a condition in which respiratory muscle strength and/or respiratory function are decreased [13]; and anal sarcopenia, associated with fecal incontinence [14]. In addition to aging, many underlying diseases are associated with the development of sarcopenia [15]; however, the state of sarcopenia is not uniform. Therefore, it is necessary to investigate the characteristics of sarcopenia according to each individual’s condition.

Therefore, we focused on the relationship between mental health and sarcopenia. Depression is a condition that interferes with activities of daily life owing to an unpleasant mood [16]. According to a 2008 World Health Organization (WHO) report, depression is the third most prevalent disease worldwide and is expected to become the most prevalent disease by 2030 [17]. Sarcopenia and depression share common pathophysiological pathways involving neurotrophins, chronic inflammation, and oxidative stress [18]. In addition, common risk factors include malnutrition [19,20], decreased activity [21,22], increased risk of falling, which causes immobility [23,24], and decreased testosterone levels [25,26]. Depression has been reported to be an independent risk factor for sarcopenia [27]. Therefore, it can be inferred that the number of older patients with sarcopenia and depressive tendencies will increase in today’s aging society.

The combination of depression and sarcopenia, a condition prevalent worldwide, may cause distinct problems that should not be underestimated. However, to the best of our knowledge, no reports have investigated the combined effects of depression and sarcopenia. Therefore, we defined sarcopenia with depression (SD). We hypothesized that patients with SD may have greater dysfunction in physical function and daily life than those with only depression (OD) or those with only sarcopenia (OS) because of the interaction between the two with similar risk factors and prognoses. This study aimed to investigate the effects of SD on older patients who require support or care.

The purpose of this study was to determine the prevalence of SD as defined by us and the characteristics of SD compared to non-depression/non-sarcopenia, OD, and OS. The results of this study are expected to contribute to our understanding of the manifestation and characteristics of depression associated with sarcopenia in older adults.

## 2. Materials and Methods

### 2.1. Study Design and Ethical Considerations

This cross-sectional study was conducted at a single center between September 2019 and September 2022. In accordance with the guidelines of the Helsinki Convention, all participants were fully informed of the purpose and measurements, and written consent was obtained from those who indicated their willingness to participate. This study was conducted after obtaining approval from the institutional ethical review committee of the International University of Health and Welfare (approval numbers 17-lo-189-7 and 21-Io-22-2).

### 2.2. Participants

The inclusion criterion was a survey of our facility (252 users) from September 2019 to September 2022. The facility involved day care for older adults undergoing rehabilitation, including pick-up and drop-off services under the long-term care insurance system. Exclusion criteria were as follows: (1) those who are at risk of falling due to stroke or balance dysfunction and have difficulty maintaining a standing position or walking (39 participants), (2) those who had difficulty in answering the questionnaire due to cognitive decline (21 participants), and (3) those with aphasia (6 participants). Excluding the 66 participants who met the exclusion criteria, 186 participants (men: 106, women: 80, mean age: 78.4 ± 8.5 years) were included in the study. None of the participants refused to participate in this study. All the study participants were older Japanese adults living in the community in need of care.

### 2.3. Basic Attributes

Sex, age, body mass index (BMI), long-term care certification level, Mini Nutritional Assessment Short-Form (MNA-sf), and comorbidities (osteoporosis, cerebrovascular disease, dementia, respiratory disease, circulatory disease, hypertension, cancer, diabetes, orthopedic disease, neurological incurable disease) were collected from the medical records.

MNA-sf was used to assess nutritional status. It consists of six survey items: (ⅰ) food intake, (ⅱ) weight loss, (ⅲ) mobility, (ⅳ) psychological stress and acute disease, (ⅴ) neuropsychological problems, and (ⅵ) calf circumference, making it useful for nutritional screening of aging adults, with higher scores indicating better nutritional status [28]. The MNA-sf evaluates nutritional status on a 14-point scale, with 11 points or higher classified as “good nutritional status”, 8 to 11 points as “nutritional status at risk”, and 7 points or lower as “low nutritional status” [29]. 

This study used the level of care required by the Japanese long-term care insurance system to assess the severity of daily functioning. The long-term care certification level is classified according to the severity of the disability into Support 1 and 2 (level requiring support in daily living) and Care 1–5 (level requiring assistance in daily living); the higher the number, the more care is required [30]. In this study, the levels of support needed (1) and (2) and care needed 1–5 were indicated on a 7-point ordinal scale. The long-term care certification level was determined by a computerized primary judgment based on a survey by a certification survey committee member, a 74-item questionnaire regarding daily living, and a physician’s opinion. Subsequently, the long-term care certification level is determined in a secondary judgment by the Long-Term Care Accreditation Board, which is composed of specialists from various fields [30]. In other words, it is an important indicator that comprehensively evaluates the living conditions and care burden of older adults from multiple perspectives.

### 2.4. Sarcopenia Assessment

The Asian Working Group for sarcopenia 2019 (AWGS 2019) algorithm [31] was used to diagnose sarcopenia. The AWGS 2019 algorithm measures grip strength, walking speed, and the Skeletal Muscle mass index (SMI) to determine the presence or absence of sarcopenia according to the reference values established for each measurement method and sex. The reference values were as follows: grip strength, <28 kg for men and <18 kg for women; usual gait speed, < 1.0 m/s; SMI, < 7.0 kg/m^2^ for men and < 5.7 kg/m^2^ for women [31]. Participants who met the criteria of both low handgrip strength and/or low gait speed and low muscle mass were considered to have sarcopenia [31].

Grip strength was measured using a digital grip strength meter (Digital Grip Strength Tester Grip D-TKK5401; Takei Kiki Kogyo, Niigata, Japan). The unit of measurement of the grip strength meter used was kg. The maximum value of the two measurements—taken twice on each side in the chair sitting position with both upper limbs hanging down to the body side—was used as the representative value. SMI was calculated by dividing the skeletal muscle mass of the limbs by the square of the participant’s height [31]. The skeletal muscle mass of the limbs was measured using the bioimpedance analysis (BIA) method with a body composition analyzer (InBody 520, InBody Japan Co., Ltd., Seoul, Korea). The results were divided by the square of the participant’s height. Walking speed was used as a measure of physical function. The walking speed was measured with the participants walking at a comfortable speed on an 11 m straight walkway consisting of a 5 m measurement section and a 3 m preliminary section set at both ends of the measurement section to avoid the effects of acceleration and deceleration. Based on the method recommended by AWGS 2019 [31], we measured the gait speed twice and used the average as the representative value. Those who needed walking aids used the same aids they relied upon in their daily lives, and all walking speeds were measured in a position where a physical therapist could immediately assist the patients, taking the risk of falling into consideration.

### 2.5. Depression Assessment

The Geriatric Depression Scale 15 (GDS-15) was used to assess depression. It consists of 15 short, simple questions about mental health, all of which can be answered with a yes/no response. A higher score indicates a tendency toward depression, and a score of 5 or higher indicates a depressive tendency [32]. The GDS-15 has proven to be the most reliable and valid for screening depression in older individuals [33], and it can be used for people with mild dementia [34]. Therefore, the GDS-15 was used in this study to screen for depression. Additionally, the GDS-15 has high internal consistency with a Cronbach’s alpha coefficient of 0.83 [35]. The GDS-15 questions are shown in Table 1. All surveys were conducted by physical or occupational therapists at our facilities.

### 2.6. Definition of Sarcopenia with Depression

In this study, sarcopenia was determined by the AWGS 2019 flow, depression was determined by a GDS-15 score of 5 or higher, and SD was defined as the co-occurrence of sarcopenia and depression.

### 2.7. Grouping

Participants were classified into four groups: control (CO: no sarcopenia, GDS-15 ≤ 5), OD (no sarcopenia, GDS-15 > 5), OS group (with sarcopenia, GDS-15 ≤ 5), and SD group (with sarcopenia, GDS-15 > 5) (Figure 1).

### 2.8. Analysis

The median, mean, and number of cases for the four study groups (sex, age, BMI, MNA-sf, long-term care certification level, grip strength, walking speed, SMI, GDS-15, and comorbidities) were calculated, and significant differences were examined using the Kruskal–Wallis test or χ^2^ test. Next, each survey item was analyzed using Spearman’s rank correlation coefficient to verify the correlation. 

Multivariate analysis was performed using the dependent variable as the survey item that showed a significant difference between the groups, with OD, OS, and SD as independent variables. Multiple regression analysis was used for grip strength, walking speed, SMI, and BMI, and binomial logistic regression analysis was used for MNA-sf and long-term care certification levels.

In the binomial logistic regression analysis, MNA-sf was classified into two categories: well nourished (MNA-sf score > 11) and undernourished or at risk of undernourishment (MNA-sf score ≤ 11). The long-term care certification level was classified into support and long-term care certification levels. Furthermore, OS and SD findings were validated by univariate and multivariate analyses. All the multivariate analyses were adjusted for age and sex. SPSS Statistics version 22 (IBM) was used for the statistical analysis, and the significance level was set at *p* < 0.05. Power analyses were performed using G*Power, version 3.1.

## 3. Results

In the overall analysis (n = 186), 41 (22.0%) patients had CO, 37 (19.9%) had OD, 50 (26.9%) had OS, and 58 (31.2%) had SD. Of the 108 patients with sarcopenia who required support or nursing care, 53.7% (men, 53.6%; women, 53.8%) had SD.

A comparison of the four groups showed significant differences in BMI, MNA-sf, long-term care certification level, grip strength, walking speed, SMI, and GDS-15. (Table 2). 

In terms of correlations among the survey items, significant correlations were found for all MNA-sf items, and significant correlations were found for all items except GDS-15 for the level of care required (Table 3).

In the multivariate analysis (adjusted variables: sex and age), SD was more strongly associated than OD and OS with worsening grip strength, gait speed, SMI, BMI, nutritional status, and long-term care certification level. In analyses regarding sex, similar associations were found (Table 4, Figure 2). 

The results of the OS vs. SD analysis are presented in Table 5 and Table 6. MNA-sf and grip strength were significantly lower in the univariate analysis of survey item OS vs. SD, and worse grip strength and MNA-sf were extracted as independent risk factors in multivariate analysis. A post-hoc analysis was performed for power analysis. Specifically, for linear multiple regression analysis using the F-test, the power was set at 0.99 based on an effect size of 0.25 (Nagelkerke R^2^ = 0.201), which was calculated from the regression equation (Table 6). The α-error probability was set at 0.05.

## 4. Discussion

Previous studies on the association between depression and sarcopenia have reported significantly higher rates of depression in older adults with sarcopenia [36] and low nutrition as a confounding factor between the two [37]. However, no study has investigated the combined effects of sarcopenia and depression. Therefore, the present study is the first to compare the characteristics of older adults with sarcopenia and depression with those of adults with sarcopenia only in terms of physical, nutritional, and lifestyle functions. This is an important report for a more detailed understanding of the pathogenesis of sarcopenia, as it presents the impact of the growing combination of both conditions worldwide.

The proportion of SD among community-dwelling older patients requiring support or care, as indicated by the flow of this study, was 31.2% of the total, and more than half of the patients with sarcopenia had depressive tendencies. The rate of depression (GDS score > 5) in patients with sarcopenia reported in a previous study of the general older population was 29.8% [38]. Therefore, a larger proportion of older patients with sarcopenia requiring care may have more SD than the general older population.

A notable point of this study is that SD showed a greater tendency to decline than OD or OS in terms of the level of care required. The level of care required is an important indicator that comprehensively evaluates the living conditions and care burden of older adults from multiple perspectives. Worsening levels of the care required are associated with life expectancy [39] and increased family care burden [40]. Factors associated with the worsening levels of the care required include decreased mental function and grip strength [41], a history of depression, subjective health perspective, decreased sense of purpose in life [42], and low nutritional status [43]. These results suggest that SD who possess both factors associated with worsening nursing care may have a more negative impact on nursing care. 

Physical function and nutritional status also tended to be worse in the SD group than in the OS or OD groups. The risk factors for depression and sarcopenia are similar in some respects, such as lack of exercise, increased inflammatory factors, and hormonal disorders of the hypothalamus–pituitary–adrenal system [27], and their co-occurrence may have a synergistic negative effect on physical function and nutritional status [27]. Previous studies have reported an association between depression, malnutrition, and grip strength [44], and the results of this study support these previous studies. The above results suggest that SD may have a greater impact than OD or OS on the degree of care required, physical function, and worsening nutritional status.

This study has several limitations. First, this was a single-center survey in Japan, the number of participants was small, and their demographics were limited. Therefore, the results may differ among regions with different diagnostic criteria for sarcopenia. Second, as this was a cross-sectional study, it was not possible to prove a causal relationship between depression and sarcopenia. Third, symptoms of mild dementia and depression overlap in many respects [45] and cannot be clearly determined. Therefore, older adults with mild dementia may be depressed. Fourth, nutritional assessment is limited to screening only and lacks quantitativeness. In the future, quantitative data, such as protein and serum albumin levels in blood, should be verified. Fifth, the system is not applicable to the general older population because it is designed for older people who require nursing care and have disabilities due to stroke or neurological diseases. In addition, we did not find any association with comorbidities in this study. In the future, it will be necessary to confirm the flow and risk factors associated with OS, OD, and SD.

However, the results of this study show that many sarcopenia patients with depressive tendencies are found among older individuals who require support and care in the community and that their physical function, nutritional status, and long-term care certification level are worse than those with only sarcopenia—this is a new finding. Therefore, SD may be a new geriatric giant among the overall older population. Further studies should be conducted to elucidate the risk factors and prognosis in longitudinal studies, in addition to more detailed SD specificities.

## 5. Conclusions

SD is common among older adults who require nursing care in the community and may have a greater impact on physical function, nutritional status, and lifestyle function compared to only sarcopenia. Future longitudinal studies are needed to elucidate the prognosis and risk factors of SD.

## Figures and Tables

**Figure 1 medicina-59-01095-f001:**
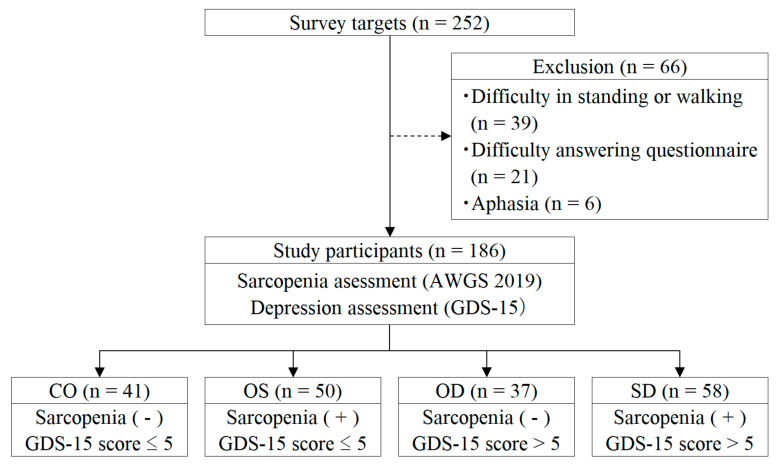
Flow diagram depicting the study participant selections. AWGS 2019: Asian Working Group for Sarcopenia 2019; GDS-15: Geriatric Depression Scale 15; CO: control; OD: only depression; OS: only sarcopenia; SD: sarcopenia with depression.

**Figure 2 medicina-59-01095-f002:**
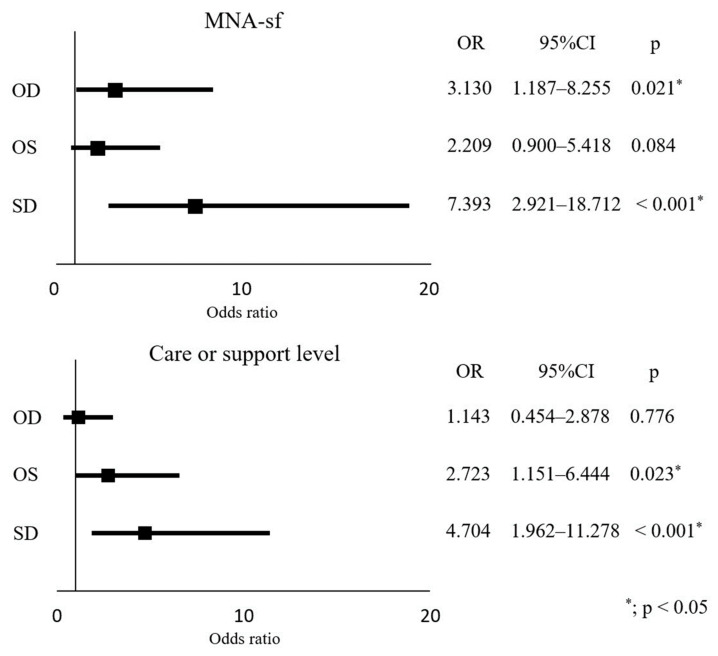
Binomial logistic regression analysis results with MNA-sf and care or support level as dependent variables. Adjusted: sex, age. OD: only depression. OS: only sarcopenia. SD: sarcopenia with depression. MNA-sf: Mini Nutritional Assessment Short-Form. OR: odds ratio. CI: confidence interval. Binomial logistic regression analysis: MNA-sf (score > 11 = 0, score ≤ 11 = 1), Care or support level (level ≤ 2 = 0, level > 2 = 1).

**Table 1 medicina-59-01095-t001:** GDS-15 Questionnaire.

1	Are you basically satisfied with your life?
2	Have you dropped many of your activities and interests?
3	Do you feel that your life is empty?
4	Do you often get bored?
5	Are you in good spirits most of the time?
6	Are you afraid that something bad is going to happen to you?
7	Do you feel happy most of the time?
8	Do you often feel helpless?
9	Do you prefer to stay at home, rather than going out and doing new things?
10	Do you feel you have more problems with memory than most?
11	Do you think it is wonderful to be alive now?
12	Do you feel pretty worthless the way you are now?
13	Do you feel full of energy?
14	Do you feel that your situation is hopeless?
15	Do you think that most people are better off than you are?

GDS-15: Geriatric Depression Scale 15.

**Table 2 medicina-59-01095-t002:** Comparison of 4 groups: control, only depression, only sarcopenia, and sarcopenia with depression.

Variables	All (n = 186)				
	(a) CO (n = 41)	(b) OD (n = 37)	(c) OS (n = 50)	(d) SD (n = 58)	*p*
Women, n (%)	21 (51.2)	20 (54.1)	18 (36.0)	21 (36.2)	0.143
Age (years)	78.0 ± 8.0	74.8 ± 9.5	79.3 ± 8.0	80.1 ± 8.0	0.067
BMI (kg/m^2^)	24.5 ± 4.7	23.3 ± 3.2	21.1 ± 3.1	20.6 ± 3.1	<0.001 *
MNA-sf (scores) ^†^	12.0 (11.0–14.0)	12.0 (10.0–13.0)	12.0 (9.25–13.0)	9.0 (6.25–11.0)	<0.001 *
Care or support level (1–7) ^†^	2.0 (1.0–3.0)	2.0 (1.0–3.0)	3.0 (1.25–3.0)	3.0 (2.0–3.75)	<0.001 *
Grip strength (kg)	23.7 ± 7.8	24.7 ± 8.9	21.4 ± 6.4	18.6 ± 6.8	<0.001 *
Gait speed (m/s)	0.8 ± 0.3	0.9 ± 0.4	0.7 ± 0.3	0.7 ± 0.3	<0.001 *
SMI (kg/m^2^)	7.1 ± 1.0	6.9 ± 1.1	5.9 ± 0.7	5.7 ± 0.8	<0.001 *
GDS-15 (scores) ^†^	2.0 (1.0–3.0)	9.0 (7.0–10.0)	3.0 (2.0–4.0)	8.0 (7.0–11.0)	<0.001 *
Comorbidities					
Osteoporosis, n (%)	5 (12.2)	2 (5.4)	5 (10.0)	7 (12.1)	0.713
Cerebrovascular disease, n (%)	19 (46.3)	20 (54.1)	23 (46.0)	30 (53.4)	0.817
Dementia, n (%)	6 (14.6)	4 (10.8)	3 (6.0)	7 (12.1)	0.583
Respiratory disease, n (%)	8 (19.5)	2 (5.4)	8 (16.0)	9 (15.5)	0.328
Circulatory disease, n (%)	7 (17.1)	5 (13.5)	15 (30.0)	17 (29.3)	0.146
Hypertension, n (%)	15 (36.6)	15 (40.5)	24 (48.0)	21 (36.2)	0.550
Cancer, n (%)	5 (12.2)	8 (21.6)	14 (28.0)	11 (19.0)	0.316
Diabetes, n (%)	4 (9.8)	9 (24.3)	13 (26.0)	13 (22.4)	0.238
Orthopedic disease, n (%)	28 (68.3)	17 (45.9)	17 (34.0)	31 (53.4)	0.134
Neurological incurable disease, n (%)	3 (7.3)	6 (16.2)	6 (12.0)	9 (15.5)	0.583

*: *p* < 0.05. †: Median (25th percentile−75th percentile). CO, control; OD, depression only. OS: only sarcopenia. SD: sarcopenia with depression. BMI: body mass index. SMI: skeletal muscle mass index. GDS-15: Geriatric Depression Scale; 15. MNA-sf: Mini Nutritional Assessment Short-Form. * *p*-value for difference between subgroups in percent (Pearson χ^2^) or median (Kruskal–Wallis test).

**Table 3 medicina-59-01095-t003:** Correlation of survey items.

	BMI	MNA-sf	Care or Support Level	Grip Strength	Gait Speed	SMI	GDS-15
BMI	-						
MNA-sf	0.434 **	-					
Care or support level	−0.156 *	−0.289 **	-				
Grip strength	0.202 **	0.297 **	−0.181 *	-			
Gait speed	0.104	0.190 **	−0.336 **	0.133	-		
SMI	0.487 **	0.285 **	−0.224 **	0.619 **	0.165 *	-	
GDS-15	−0.154 *	−0.346 **	0.124	−0.086	−0.063	−0.142	-

*: *p* < 0.05, **: *p* < 0.01. BMI: body mass index. MNA-sf: Mini Nutritional Assessment Short-Form. SMI: skeletal muscle mass index. GDS-15: Geriatric Depression Scale 15.

**Table 4 medicina-59-01095-t004:** Multivariate analysis of factors associated with OD, OS, and SD compared with CO.

All (n = 186)						
Variables	Grip Strength (kg)			Gait speed (m/s)		
	β	SE	*p*	β	SE	*p*
OD	0.031	1.265	0.639	0.125	0.067	0.154
OS	−0.197	1.175	0.004 *	−0.198	0.062	0.029 *
SD	−0.368	1.146	<0.001 *	−0.292	0.060	0.002 *
	SMI (kg/m^2^)			BMI (kg/m^2^)		
	β	SE	*p*	β	SE	*p*
OD	−0.058	0.156	0.308	−0.147	0.795	0.077
OS	−0.582	0.145	<0.001 *	−0.397	0.739	<0.001 *
SD	−0.692	0.141	<0.001 *	−0.464	0.721	<0.001 *
Men (n = 106)						
	Grip strength (kg)			Gait speed (m/s)		
	β	SE	*p*	β	SE	*p*
OD	0.020	2.134	0.851	0.080	0.099	0.498
OS	−0.278	1.849	0.018 *	−0.131	0.086	0.307
SD	−0.446	1.825	<0.001 *	−0.332	0.085	0.012 *
	SMI (kg/m^2^)			BMI (kg/m^2^)		
	β	SE	*p*	β	SE	*p*
OD	−0.115	0.242	0.176	−0.087	−0.104	0.347
OS	−0.699	0.210	<0.001 *	−0.429	−0.420	<0.001 *
SD	−0.837	0.207	<0.001 *	−0.484	−0.481	<0.001 *
Women (n = 80)						
	Grip strength (kg)			Gait speed (m/s)		
	β	SE	*p*	β	SE	*p*
OD	0.070	1.306	0.531	0.160	0.089	0.212
OS	−0.171	1.323	0.121	−0.310	0.091	0.015 *
SD	−0.451	1.272	<0.001 *	−0.275	0.087	0.032 *
	SMI (kg/m^2^)			BMI (kg/m^2^)		
	β	SE	*p*	β	SE	*p*
OD	0.002	0.193	0.984	−0.190	1.235	0.136
OS	−0.588	0.196	<0.001 *	−0.361	1.250	0.004 *
SD	−0.685	0.188	<0.001 *	−0.458	1.202	<0.001 *

*: *p* < 0.05. Adjusted: sex, age. OD: only depression. OS: only sarcopenia. SD: sarcopenia with depression. SMI: skeletal muscle mass index. BMI: body mass index. SE: standard error. OR: odds ratio. CI: confidence interval. Multiple regression analysis: grip strength, gait speed, SMI, BMI.

**Table 5 medicina-59-01095-t005:** Univariate analysis of OS vs. SD.

Variables	OS (n = 50)	SD (n = 58)	*p*
Female, n (%)	18 (36.0)	21 (36.2)	0.982
Age (years)	79.3 ± 8.0	80.1 ± 8.0	0.579
BMI (kg/m^2^)	21.1 ± 3.1	20.6 ± 3.1	0.425
MNA-sf (scores) ^†^	12.0 (9.25–13.0)	9.0 (6.25–11.0)	<0.001 *
Care or support level (1–7) ^†^	3.0 (1.25–3.0)	3.0 (2.0–3.75)	0.149
Grip strength (kg)	21.4 ± 6.4	18.6 ± 6.8	0.026 *
Gait speed (m/s)	0.7 ± 0.3	0.7 ± 0.3	0.370
SMI (kg/m^2^)	5.9 ± 0.7	5.7 ± 0.8	0.146

*: *p* < 0.05. †: median (25th percentile−75th percentile). Analysis: Mann–Whitney U test. OS: only sarcopenia. SD: sarcopenia with depression. BMI: body mass index. MNA-sf: Mini Nutritional Assessment Short-Form. SMI: skeletal muscle mass index.

**Table 6 medicina-59-01095-t006:** Multivariate analysis of OS vs. SD.

Variables	β	SE	OR	95% CI	*p*
Grip strength	−0.093	0.046	0.911	0.832–0.997	0.044 *
MNA-sf	−0.212	0.072	0.809	0.703–0.931	0.003 *

* *p* < 0.05. Analysis: binomial logistic regression analysis (OS = 0, SD = 1). Adjusted: sex, age. OS: only sarcopenia. SD: sarcopenia with depression. MNA-sf: Mini Nutritional Assessment Short-Form. SE: standard error. OR: odds ratio. CI: confidence interval.

## Data Availability

Research data are not shared.

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
