# Peer review of "Sarcopenia with Depression Presents a More Severe Disability Than Only Sarcopenia among Japanese Older Adults in Need of Long-Term Care"

_medicina, 2023, doi:10.3390/medicina59061095_

Round 1

Reviewer 1 Report

First of all, thank you for your contribution to the field.

The last sentence of the introduction is a very assertive sentence. In order to reach such a result, I think that a multicenter and a large number of cases are needed!

In the section where The geritric depression scale 15 is explained, it is not necessary to write all the questions of the scale, this section can be simplified.

Author Response

Response to Reviewer 1

Thank you very much for reviewing my manuscript.

We have revised the manuscript in accordance with your suggestions.

Q1. First of all, thank you for your contribution to the field.

A1. Thank you for your words of encouragement.

Through this study, we hope to contribute to the investigation of sarcopenia with depression.

Q2. The last sentence of the introduction is a very assertive sentence. In order to reach such a result, I think that a multicenter and a large number of cases are needed!

A2. Thank you for your comments. As indicated by you, we considered the number of participants and study facilities to be insufficient to elucidate the pathogenesis of sarcopenia with depression. Therefore, we have revised the last statement in the Introduction section as follows:

【Before: L71-72】

The results of this study are expected to help in understanding the pathophysiology of depression with sarcopenia in the older adults.

   【After: L78-79】

The results of this study are expected to contribute to our understanding of the manifestation and characteristics of depression associated with sarcopenia in older adults.  

Q3. In the section where The geriatric depression scale 15 is explained, it is not necessary to write all the questions of the scale, this section can be simplified.

A3. Thank you for your comments. As you pointed out, the original text was redundant. Therefore, we decided to express this information in a table based on the suggestions of the other reviewers. We have deleted the following text and added a table:

【Delete: L146 - 155】

The questions on the GDS-15 are as follows: 1; Are you basically satisfied with your life?, 2; Have you dropped many of your activities and interests?, 3; Do you feel that your life is empty?, 4; Do you often get bored?, 5; Are you in good spirits most of the time?, 6; Are you afraid that something bad is going to happen to you?, 7; Do you feel happy most of the time?, 8; Do you often feel helpless?, 9; Do you prefer to stay at home, rather than going out and doing new things?, 10; Do you feel you have more problems with memory than most?, 11; Do you think it is wonderful to be alive now?, 12; Do you feel pretty worthless the way you are now?, 13; Do you feel full of energy?, 14; Do you feel that your situation is hopeless?, 15; Do you think that most people are better off than you are?.

【Correction: L163】

The GDS-15 questions are shown in Table 1.

【Correction: Table1】

Table1. GDS-15 Questionnaire

1

Are you basically satisfied with your life?

2

Have you dropped many of your activities and interests?

3

Do you feel that your life is empty?

4

Do you often get bored?

5

Are you in good spirits most of the time?

6

Are you afraid that something bad is going to happen to you?

7

Do you feel happy most of the time?

8

Do you often feel helpless?

9

Do you prefer to stay at home, rather than going out and doing new things?

10

Do you feel you have more problems with memory than most?

11

Do you think it is wonderful to be alive now?

12

Do you feel pretty worthless the way you are now?

13

Do you feel full of energy?

14

Do you feel that your situation is hopeless?

15

Do you think that most people are better off than you are?

GDS-15: Geriatric depression scale 15.

Reviewer 2 Report

REVIEWER COMMENTS: 

I want to thank the authors for the opportunity to review this work. I feel it could make contributions to health care area.  However, it needs major revisions in order to be improved.

Introduction section:

1. It would be interesting to include the concept of sarcopenia and depression in the introduction section.

2. You should include within the theoretical framework references that address depression and sarcopenia, providing information on this relationship and associated risk factors.

Materials and Methods section:

3. Where are the participants from? More information on this should be included.

4. Information on the sample size calculation should be included to justify a sufficient size for the proposed data analysis.

5. Indicate the units of measurement of grip strength.

6. The formula used to calculate the Skeletal Muscle mass must be indicated, from which the index will later be obtained (“The results were divided by the square of the participant’s height”).

7. Why was the gait speed measured twice and used the average as the representative value? Generally, it is usually carried out once as a test and the second measurement is taken. Include a bibliographical reference of the protocol used.

8. How were those people who were at risk of falling assisted?

9. Based on previous bibliography, indicate an internal consistency value of the depression scale used.

10. Include more information about who carried out the physical measurements.

11. It would be better to include the items of the depression scale in a table, than by means of a text description.

12. Include in the statistical analysis section the confounding variables used in the multiple regression analysis.

Discusion section:

13. You refer to the fact that "This is the first study to examine the characteristics of patients with sarcopenia and depressive tendencies". It would be interesting to review the previous available evidence showing the trend in these patients. Based on these previous studies, the discussion should be established in greater depth.

For example see among others studies:

Delibaş DH, Eşkut N, İlhan B, et al. Clarifying the relationship between sarcopenia and depression in geriatric outpatients. Aging Male. 2021;24(1):29-36. doi:10.1080/13685538.2021.1936482

Kitamura A, Seino S, Abe T, et al. Sarcopenia: prevalence, associated factors, and the risk of mortality and disability in Japanese older adults.J Cachexia Sarcopenia Muscle. 2021;12(1):30-38. doi:10.1002/jcsm.12651

Conclusions section:

14. Conclusions should be more specific by highlighting clear take-home messages.

15. The last sentence is repeated twice, check the wording.

Author Response

Response to Reviewer 2

Thank you very much for reviewing my manuscript.

I have revised the manuscript in accordance with your suggestions.

Q1. I want to thank the authors for the opportunity to review this work. I feel it could make contributions to health care area.  However, it needs major revisions in order to be improved.

A1. Thank you for reviewing this manuscript.

We have revised the paper according to your comments and suggestions and would appreciate your review.

Q2. It would be interesting to include the concept of sarcopenia and depression in the introduction section.

A2. Thank you for your comments. As indicated by you, we believed that including the concepts of sarcopenia and depression in the Introduction section would make the paper more interesting to the readers. Please find the additions made based on your comments below.

  【Correction: L41-42】

Sarcopenia is characterized by the progressive and generalized loss of skeletal muscle mass and strength with age and disease.

【Correction: L57-59】

Depression is a condition that interferes with activities of daily life owing to an unpleasant mood.

Q3. You should include within the theoretical framework references that address depression and sarcopenia, providing information on this relationship and associated risk factors.

A3. Thank you for the suggestion. we have included information on the relationship between depression and sarcopenia in the Introduction section of the manuscript with reference to your suggestion.

  【Correction: L61-65】

Sarcopenia and depression share common pathophysiological pathways involving neurotrophins, chronic inflammation, and oxidative stress (16). In addition, common risk factors include malnutrition (17,18), decreased activity (19,20), increased risk of falling, which causes immobility (21,22), and decreased testosterone levels (23,24).   

【Correction: L356-375】

16)           Chen GQ, Wang GP, Lian Y. Relationships Between Depressive Symptoms, Dietary Inflammatory Potential, and Sarcopenia: Mediation Analyses. Front Nutr. 2022 Feb 17; 9: 844917. doi: 10.3389/fnut.2022.844917. PMID: 35252313; PMCID: PMC8891449.

17)           Wei J, Fan L, Zhang Y, Li S, Partridge J, Claytor L, Sulo S. Association Between Malnutrition and Depression Among Community-Dwelling Older Chinese Adults. Asia Pac J Public Health. 2018 Mar;30(2):107-117. doi: 10.1177/1010539518760632. Epub 2018 Mar 7. PMID: 29514465.

18)           Sieber CC. Malnutrition and sarcopenia. Aging Clin Exp Res. 2019 Jun;31(6):793-798. doi: 10.1007/s40520-019-01170-1. Epub 2019 May 30. PMID: 31148100.

19)           Veronese N, Stubbs B, Trevisan C, Bolzetta F, De Rui M, Solmi M, Sartori L, Musacchio E, Zambon S, Perissinotto E, Baggio G, Crepaldi G, Manzato E, Maggi S, Sergi G. Poor Physical Performance Predicts Future Onset of Depression in Elderly People: Progetto Veneto Anziani Longitudinal Study. Phys Ther. 2017 Jun 1;97(6):659-668. doi: 10.1093/ptj/pzx017. PMID: 28201628.

20)           Ribeiro Santos V, Dias Correa B, De Souza Pereira CG, Alberto Gobbo L. Physical Activity Decreases the Risk of Sarcopenia and Sarcopenic Obesity in Older Adults with the Incidence of Clinical Factors: 24-Month Prospective Study. Exp Aging Res. 2020 Mar-Apr;46(2):166-177. doi: 10.1080/0361073X.2020.1716156. Epub 2020 Jan 23. PMID: 31971091.

21)           Stubbs B, Stubbs J, Gnanaraj SD, Soundy A. Falls in older adults with major depressive disorder (MDD): a systematic review and exploratory meta-analysis of prospective studies. Int Psychogeriatr. 2016 Jan;28(1):23-9. doi: 10.1017/S104161021500126X. Epub 2015 Aug 3. PMID: 26234532.

22)           Yeung SSY, Reijnierse EM, Pham VK, Trappenburg MC, Lim WK, Meskers CGM, Maier AB. Sarcopenia and its association with falls and fractures in older adults: A systematic review and meta-analysis. J Cachexia Sarcopenia Muscle. 2019 Jun;10(3):485-500. doi: 10.1002/jcsm.12411. Epub 2019 Apr 16. PMID: 30993881; PMCID: PMC6596401.

23)           Sgrò P, Sansone M, Sansone A, Sabatini S, Borrione P, Romanelli F, Di Luigi L. Physical exercise, nutrition and hormones: three pillars to fight sarcopenia. Aging Male. 2019 Jun;22(2):75-88. doi: 10.1080/13685538.2018.1439004. Epub 2018 Feb 16. PMID: 29451419.

24)           Zarrouf FA, Artz S, Griffith J, Sirbu C, Kommor M. Testosterone and depression: systematic review and meta-analysis. J Psychiatr Pract. 2009 Jul;15(4):289-305. doi: 10.1097/01.pra.0000358315.88931.fc. PMID: 19625884.

Q4. Where are the participants from? More information on this should be included.

A4. Thank you for your consideration of this manuscript. All participants in this study were Japanese. We have made the following revisions in response to your comments:

【Correction: L99-100】

All the study participants were older Japanese adults living in the community in need of

care.

Q5. Information on the sample size calculation should be included to justify a sufficient size for the proposed data analysis.

A5. Thank you for pointing this out. G*Power was used to calculate the sample size. We have added the following text to reflect on your comments in the manuscript.

【Correction: L194-195】

Power analyses were performed using G*Power, version 3.1.

【Correction: L233-236】

A post- hoc analysis was performed for power analysis. Specifically, for linear multiple regression analysis using the F-test, the power was set at 0.99 based on an effect size of 0.25 (Nagelkerke R2 = 0.201), which was calculated from the regression equation (Table 6).

Q6. Indicate the units of measurement of grip strength.

A6. Thank you for the suggestion. We have made the following corrections in this manuscript to reflect your suggestions:

【Correction: L138-139】

The unit of measurement of the grip strength meter used was kg.

Q7. The formula used to calculate the Skeletal Muscle mass must be indicated, from which the index will later be obtained (“The results were divided by the square of the participant’s height”).

A7. Thank you for your suggestion. We felt that the wording of the text on SMI calculation method was inadequate. We have made the following revisions to reflect your comments in the manuscript.

  【Before: L132-135】

  SMI was measured using the bioimpedance analysis (BIA) method with a body composition analyzer (InBody520, InBody Japan Co., Ltd., Seoul, Korea). The results were divided by the square of the participant’s height.

    【After: 141-144】

    SMI was calculated by dividing the skeletal muscle mass of the limbs by the square of the participant’s height (29). The skeletal muscle mass of the limbs was measured using the bioimpedance analysis (BIA) method with a body composition analyzer (InBody 520, InBody Japan Co., Ltd., Seoul, Korea).

Q8. Why was the gait speed measured twice and used the average as the representative value? Generally, it is usually carried out once as a test and the second measurement is taken. Include a bibliographical reference of the protocol used.

A8. Thank you for the suggestion. We have taken your instructions into consideration and have included the protocol used to measure gait speed as a citation.

【Before: L139-140】

    We measured the gait speed twice and used the average as the representative value.

    【After: L149-151】

Based on the method recommended by AWGS 2019 (29), we measured the gait speed twice and used the average as the representative value.

Q9. How were those people who were at risk of falling assisted?

A9. Thank you for your question. We believe that this was misleading. We excluded patients who were at risk of falling. However, in the event of a fall, a physical therapist observed the patients in a position where they could be immediately assisted. Therefore, the following text has been modified:

  【Before: L140-141】

  Those who needed walking aids used the walking aids they normally used in their daily lives, and those who were at risk of falling provided minimal assistance.

  【After: L151-154】

Those who needed walking aids used the same aids they relied upon in their daily lives, and all walking speeds were measured in a position where a physical therapist could immediately assist the patients, taking the risk of falling into consideration.

Q10. Based on previous bibliography, indicate an internal consistency value of the depression scale used

A10. Thank you for the suggestion. The GDS-15 used in this study has high internal consistency, with a Cronbach’s alpha reliability coefficient of 0.83. In response to your suggestion, we have added the following information to the manuscript:

【Correction: L162-163】

Additionally, the GDS-15 has high internal consistency with a Cronbach’s alpha coefficient of 0.83 (33).

【Correction: L394-395】

33)   Sugishita K, Sugishita M, Hemmi I, Asada T, Tanigawa T. A Validity and Reliability Study of the Japanese Version of the Geriatric Depression Scale 15 (GDS-15-J). Clin Gerontol. 2017 Jul-Sep;40(4):233-240. doi: 10.1080/07317115.2016.1199452. Epub 2016 Jun 29. Erratum in: Clin Gerontol. 2018 Jan-Feb;41(1):108. PMID: 28452641.

Q11. Include more information about who carried out the physical measurements

A11. Thank you for your question. Measurements were performed by physical or occupational therapists at our facility. To answer your question, we have added the following information to the manuscript:

 【Correction: L163-164】

All surveys were conducted by physical or occupational therapists at our facilities.

Q12. It would be better to include the items of the depression scale in a table, than by means of a text description.

A12. Thank you for the suggestion. We have prepared a table that reflects your suggestions.

【Correction: Table1】

Table1. GDS-15 Questionnaire

1

Are you basically satisfied with your life?

2

Have you dropped many of your activities and interests?

3

Do you feel that your life is empty?

4

Do you often get bored?

5

Are you in good spirits most of the time?

6

Are you afraid that something bad is going to happen to you?

7

Do you feel happy most of the time?

8

Do you often feel helpless?

9

Do you prefer to stay at home, rather than going out and doing new things?

10

Do you feel you have more problems with memory than most?

11

Do you think it is wonderful to be alive now?

12

Do you feel pretty worthless the way you are now?

13

Do you feel full of energy?

14

Do you feel that your situation is hopeless?

15

Do you think that most people are better off than you are?

GDS-15: Geriatric depression scale 15.

Q13. Include in the statistical analysis section the confounding variables used in the multiple regression analysis.

A13. Thank you for your comments. We have added the following to reflect on your comments regarding this paper.

【Correction: L192-193】

All the multivariate analyses were adjusted for age and sex.

Q14. You refer to the fact that "This is the first study to examine the characteristics of patients with sarcopenia and depressive tendencies". It would be interesting to review the previous available evidence showing the trend in these patients. Based on these previous studies, the discussion should be established in greater depth.

For example, see among others studies:

Delibaş DH, Eşkut N, İlhan B, et al. Clarifying the relationship between sarcopenia and depression in geriatric outpatients. Aging Male. 2021;24(1):29-36. doi:10.1080/13685538.2021.1936482

Kitamura A, Seino S, Abe T, et al. Sarcopenia: prevalence, associated factors, and the risk of mortality and disability in Japanese older adults.J Cachexia Sarcopenia Muscle. 2021;12(1):30-38. doi:10.1002/jcsm.12651

A14. Thank you for your comments. We believe that the differences from previous studies were not easy to understand and have accordingly revised the wording of the manuscript to make it more comprehensive. We have revised the article by adding the previous studies indicated by you to present a more detailed discussion.

【Before: L238-239】

This is the first study to examine the characteristics of patients with sarcopenia and depressive tendencies.

【After: L246-252】

Previous studies on the association between depression and sarcopenia have reported significantly higher rates of depression in older adults with sarcopenia (34) and low nutrition as a confounding factor between the two (35). However, no study has investigated the combined effects of sarcopenia and depression. Therefore, the present study is the first to compare the characteristics of older adults with sarcopenia and depression with those of adults with sarcopenia only, in terms of physical, nutritional, and lifestyle functions.

【Correction: L396-400】

34) Kitamura A, Seino S, Abe T, Nofuji Y, Yokoyama Y, Amano H, Nishi M, Taniguchi Y, Narita M, Fujiwara Y, Shinkai S. Sarcopenia: prevalence, associated factors, and the risk of mortality and disability in Japanese older adults. J Cachexia Sarcopenia Muscle. 2021 Feb;12(1):30-38. doi: 10.1002/jcsm.12651. Epub 2020 Nov 25. PMID: 33241660; PMCID: PMC7890144.

35) Delibaş DH, Eşkut N, İlhan B, Erdoğan E, Top Kartı D, Yılmaz Küsbeci Ö, Bahat G. Clarifying the relationship between sarcopenia and depression in geriatric outpatients. Aging Male. 2021 Dec;24(1):29-36. doi: 10.1080/13685538.2021.1936482. PMID: 34151708.

Q15. Conclusions should be more specific by highlighting clear take-home messages.

Q16. The last sentence is repeated twice, check the wording.

A15,16. Thank you for the suggestion. We have revised our Conclusions to reflect your comments as follows: 

【Before: L288-294】

SD is common among older individuals living in the community. Patients with SD require support and care, and the condition has a greater impact on physical function, nutritional status, and decline in life function compared to OD and OS. Therefore, it is desirable to elucidate the process leading to SD and investigate the risk factors and prognosis. Although the number of participants in this study is small, it is hoped that sarcopenia with depression will be investigated worldwide in the future.

【After: L301-304】

SD is common among older adults who require nursing care in the community and may have a greater impact on physical function, nutritional status, and lifestyle function compared to that by only sarcopenia. Future longitudinal studies are needed to elucidate the prognosis and risk factors of SD.

Round 2

Reviewer 2 Report

-The definitions used for sarcopenia and depression must be justified by bibliographic reference.

-The justification of the sample size must be indicated not only with the name of the program used, but also with the alpha adjustments, effect size, etc. Describe these data and indicate the minimum n.

Author Response

Response to Reviewer 2

Thank you very much for reviewing my manuscript.

I have revised the manuscript in accordance with your suggestions.

Q1. The definitions used for sarcopenia and depression must be justified by bibliographic reference.

A1. Thank you for pointing this out.

We have added the cited references to reflect your guidance.

【Correction: L42】

    Sarcopenia is characterized by the progressive and generalized loss of skeletal muscle mass and strength with age and disease (1).

【Correction: L59】

Depression is a condition that interferes with activities of daily life owing to an un-pleasant mood (16).

【Correction: L324-325】

1) Cruz-Jentoft AJ, Sayer AA. Sarcopenia. Lancet. 2019 Jun 29;393(10191):2636-2646. doi: 10.1016/S0140-6736(19)31138-9. Epub 2019 Jun 3. Erratum in: Lancet. 2019 Jun 29;393(10191):2590. PMID: 31171417.

【Correction: L357-358】

16) Cui R. Editorial: A Systematic Review of Depression. Curr Neuropharmacol. 2015;13(4):480. doi: 10.2174/1570159x1304150831123535. PMID: 26412067; PMCID: PMC4790400.

Q2. The justification of the sample size must be indicated not only with the name of the program used, but also with the alpha adjustments, effect size, etc. Describe these data and indicate the minimum n.

A2. Thank you for pointing this out.

As the power analysis was conducted after the suggestion was made, a post-hoc method was used in this study. Therefore, the minimum number of n cannot be calculated because the power is calculated from the number of n participants in this study. The effect size is 0.25, which is calculated from the Nagelkerke R2 in the regression equation. As you pointed out, no information on α exists; thus, the following additions have been made: the effect size, α-error probability, and minimum n as follows:

【Correction: L236】

A post-hoc analysis was performed for power analysis. Specifically, for linear multiple regression analysis using the F-test, the power was set at 0.99 based on an effect size of 0.25 (Nagelkerke R2 = 0.201), which was calculated from the regression equation (Table 6). The α-error prob was set at 0.05.